# Effects of Candidalysin Derived from *Candida albicans* on the Expression of Pro-Inflammatory Mediators in Human Gingival Fibroblasts

**DOI:** 10.3390/ijms24043256

**Published:** 2023-02-07

**Authors:** Yasufumi Nishikawa, Yoritoki Tomotake, Hiromichi Kawano, Koji Naruishi, Jun-ichi Kido, Yuka Hiroshima, Akikazu Murakami, Tetsuo Ichikawa, Hiromichi Yumoto

**Affiliations:** 1Oral Implant Center, Tokushima University Hospital, 2-50-1 Kuramoto-cho, Tokushima 770-8503, Japan; 2Department of Periodontology and Endodontology, Institute of Biomedical Sciences, Tokushima University Graduate School, 3-18-15 Kuramoto-cho, Tokushima 770-8504, Japan; 3Department of Oral Microbiology, Institute of Biomedical Sciences, Tokushima University Graduate School, 3-18-15 Kuramoto-cho, Tokushima 770-8504, Japan; 4Department of Prosthodontics and Oral Rehabilitation, Institute of Biomedical Sciences, Tokushima University Graduate School, 3-18-15 Kuramoto-cho, Tokushima 770-8504, Japan

**Keywords:** peri-implantitis, *Candia albicans*, candidalysin, sIL-6R, human gingival fibroblasts

## Abstract

*Candida albicans* (*Ca*) is frequently detected in the peri-implant sulcus with peri-implantitis, a major postoperative complication after oral implant therapy. However, the involvement of *Ca* in the pathogenesis of peri-implantitis remains unclear. In this study, we aimed to clarify *Ca* prevalence in the peri-implant sulcus and investigated the effects of candidalysin (Clys), a toxin produced by *Ca*, on human gingival fibroblasts (HGFs). Peri-implant crevicular fluid (PICF) was cultured using CHROMagar and *Ca* colonization rate and colony numbers were calculated. The levels of interleukin (IL)-1β and soluble IL-6 receptor (sIL-6R) in PICF were quantified by enzyme-linked immunosorbent assay (ELISA). Pro-inflammatory mediator production and intracellular signaling pathway (MAPK) activation in HGFs were measured by ELISA and Western blotting, respectively. The *Ca* colonization rate and the average number of colonies in the peri-implantitis group tended to be higher than those in the healthy group. IL-1β and sIL-6R levels in the PICF were significantly higher in the peri-implantitis group than in the healthy group. Clys significantly induced IL-6 and pro-matrix metalloproteinase (MMP)-1 productions in HGFs, and co-stimulation with Clys and sIL-6R increased IL-6, pro-MMP-1, and IL-8 production levels in HGFs compared with Clys stimulation alone. These findings suggest that Clys from *Ca* plays a role in the pathogenesis of peri-implantitis by inducing pro-inflammatory mediators.

## 1. Introduction

Oral implant therapy has already been established as a useful prosthetic treatment to restore dentition and reestablish the masticatory function of partially or fully edentulous patients with high predictability and survival rates [1]. Defined clinical and histological characteristics of healthy peri-implant tissues allow the recognition of diseases [2]. The peri-implant mucosa is primarily composed of epithelial cells and fibroblasts. In particular, fibroblasts produce extracellular matrices, such as collagen fiber, for wound healing, reconstitution of tissues degraded by matrix metalloproteinases (MMPs), and maintenance of homeostasis [3]. However, it has been reported that peri-implant diseases, including peri-implant mucositis and peri-implantitis, are caused at a certain rate as biological complications after oral implant therapy. A multicenter retrospective study examining the functions of implants for more than 3 years reported that the incidence rate of peri-implantitis was 39.7% (23.9% and 15.8% incidences of peri-mucositis and peri-implantitis, respectively) [4]. A recent systematic review and meta-analysis reported that the prevalence of peri-implantitis was 19.53% at the patient level and 12.53% at the implant level [5]. Peri-implantitis is characterized by inflammation of the peri-implant connective tissue and progressive loss of supporting bone. The development of peri-implantitis is related to a history of periodontal disease, poor oral hygiene, and a lack of regular maintenance following oral implant therapy. A previous systematic review reported that the microbiological characteristics of peri-implantitis and periodontitis are different and that the quantity rather than the quality of the microorganisms, is the determinant in the development of peri-implantitis [6]. Characteristically, *Candida albicans* (*Ca*) has been revealed to be frequently detected in peri-implantitis sites compared with healthy peri-implant sites [7,8]. Candida species are generally harmless commensal fungi that reside in the digestive, genital, oral, and skin tissues of humans. However, under certain circumstances, such as inadequate antibiotic application, stress, immunosuppressive therapy, and changes in nutritional composition, *Ca* can cause systemic candidiasis in particular, as well as a superficial mucosal disease [9]. *Ca* is also involved in invasive candidiasis, including candidemia and deep infections such as intra-abdominal abscess, peritonitis, and osteomyelitis [10]. Under conditions of high temperature, hypoxia, and starvation, this fungus switches from yeast (commensal form) to hypha (pathogenic form), which can invade the substrate of organisms [11]. Candidalysin (Clys), a hyphae-associated peptide toxin secreted from *Ca*, can cause intracellular calcium influx by forming pores in the plasma membrane of epithelial cells and following the release of epidermal growth factor receptor (EGFR) ligand, activation of EGFR, phosphorylation of p38, extracellular signal-regulated protein kinase (ERK), and activation of c-Fos, which finally results in the release of pro-inflammatory mediators [12,13]. It has also been reported that Clys is essential for the translocation of *Ca* via the intestine and causes systemic infections [14]. Moreover, another study reported a rich genetic diversity of *Ca* strains in the intestines of patients with inflammatory bowel disease (IBD) and the promotion of Clys-dependent inflammation [15]. However, regarding the pathogenesis of peri-implantitis, the precise mechanism by which Clys toxins from *Ca* affect the progression of peri-implantitis remains unclear.

The level of chemical mediators, such as interleukin (IL)-1β, tumor necrosis factor (TNF)-α, and calprotectin, in peri-implant crevicular fluid (PICF) correlates with the progression of peri-implantitis and its usefulness as a biomarker of peri-implant disease is becoming clear [16,17]. Our previous studies have shown that the concentration of the soluble form of the IL-6 receptor (sIL-6R), an IL-6 agonist released by lymphocytes and macrophages, in the gingival crevicular fluid (GCF) was significantly higher in periodontitis sites than in healthy sites [18]. Previous studies have shown that IL-6 induces the production of MMP-1, vascular endothelial growth factor (VEGF), monocyte chemoattractant protein (MCP)-1, and basic fibroblast growth factor (bFGF) in the presence of sIL-6R in human gingival fibroblasts (HGFs) [19]. However, to date, there have been no reports on sIL-6R in the PICF of peri-implantitis sites, and its involvement in the pathogenesis of peri-implantitis is unknown. 

In this study, we have investigated the colonization rate and the average number of colonies of *Ca* and sIL-6R levels in the PICF of peri-implantitis sites to clarify the effects of Clys on the inflammatory reaction in HGFs with regard to the pathogenesis of peri-implantitis.

## 2. Results

### 2.1. Clinical Study

This cross-sectional study included 19 patients who received implant therapy (healthy, 9 subjects; peri-implantitis, 10 subjects). We investigated the clinical oral status using probing pocket depth (PPD), modified plaque index (mPI), and modified gingival index (mGI), as well as the presence and colonization rate of *Ca* in 19 patients (Table 1), and quantified the levels of inflammatory parameters in PICF (Figure 1). There were significant differences in PPD, mPI score, mGI score, and IL-1β and sIL-6R levels between the two groups. The colonization rate of *Ca* and the average number of colonies in the peri-implantitis group were higher than those in the healthy group.

### 2.2. In Vitro Study

#### 2.2.1. Cell Cytotoxicity of Clys to HGFs

We first assessed the cytotoxicity of Clys on HGFs using a lactate dehydrogenase (LDH) assay. LDH levels were significantly higher 24 and 48 h after stimulation with 10 μM Clys. Based on these results, the Clys concentration was set to 5 μM for subsequent experiments (Figure 2).

#### 2.2.2. Production Levels of Inflammation-Related Molecules in HGFs Stimulated with Clys

To determine the role of Clys in the production of inflammation-related molecules from HGFs, HGFs were stimulated with various concentrations of Clys. The production levels of IL-6 and pro-MMP-1 in HGFs stimulated with Clys for 24 h significantly increased in a concentration-dependent manner (Figure 3A,B). The levels of MCP-1, IL-8, and tissue inhibitor of metalloproteinase (TIMP)-1 did not change (Figure 3C–E). The ratio of pro-MMP-1/TIMP-1 also significantly increased in a concentration-dependent manner (Figure 3F).

#### 2.2.3. Clys-Induced Intracellular Signaling Pathways, MAPKs, in HGFs

Next, to elucidate the mechanisms of the pro-inflammatory response of HGFs in response to Clys, we investigated the intracellular signal transduction pathways in Clys-stimulated HGFs. ERK and p38 were phosphorylated in HGFs stimulated with Clys in a concentration-dependent manner, whereas c-Jun N-terminal kinase (JNK) was not phosphorylated (Figure 4A). The production levels of IL-6 and pro-MMP-1 were significantly decreased by the p38-specific inhibitor SB2013580 and the ERK-specific inhibitor U0126, respectively (Figure 4B).

#### 2.2.4. Effects of sIL-6R on Clys-Induced Inflammation-Related Molecules Production and Activation of Glycoprotein (gp)130/Signal Transducer and Activator of Transcription (Stat)3

This clinical study showed that sIL-6R levels in the PICF were significantly higher in the peri-implantitis group than in the healthy group (Figure 1). Next, we investigated the effects of sIL-6R on Clys-induced inflammation-related molecule production and its signal transduction pathway. When sIL-6R was applied for 10 min to HGFs after stimulation with Clys for 24 or 48 h, glycoprotein (gp)130 and signal transducer and activator of transcription (Stat)3 were significantly phosphorylated (Figure 5A). The production levels of IL-6, pro-MMP-1, and IL-8 at 48 h in HGFs co-stimulated with Clys and sIL-6R were significantly higher than those in HGFs stimulated with Clys only, but the production levels of TIMP-1 and MCP-1 did not change even when HGFs were co-stimulated with Clys and sIL-6R (Figure 5B).

## 3. Discussion

Oral implant therapy has become an established option for prosthetic treatment owing to its high predictability. The number of patients who receive oral implant therapy is increasing, but both short- and long-term failures have been reported [20]. Clinical features of peri-implantitis and periodontitis are common; however, significant histopathological differences exist between these lesions [21]. Previous studies have shown that peri-implantitis lesions are more than twice as large as periodontitis lesions and have significantly greater proportions, numbers, and densities of immune cells [22]. Peri-implantitis is an irreversible disease; therefore, it is particularly important to reveal the pathogenesis of peri-implantitis and to develop a treatment for this disease. de Mendoza et al. [23] reported a larger presence of Candida species (especially *Ca*) in PICF collected from peri-implantitis sites compared with healthy implant sites. *Ca* produces secreted aspartyl proteinases (SAPs) and phospholipases (PLs), causing damage to host cells and tissues [9]. In addition, Moyes et al. reported that *Ca* disrupts the epithelial membranes by producing peptide toxins (Candidalysin: Clys) [12]. Canabarro et al. [24] reported that subgingival colony formation in some yeasts, particularly *Ca*, is associated with the severity of chronic periodontitis. However, the involvement of *Ca* in the progression of peri-implantitis remains unknown. 

We previously reported that sIL-6R was detected at higher levels in the GCF of periodontitis sites than in healthy sites [18] and that these molecules induce MMP-1 expression in the secondary response to IL-6 produced by IL-1β and indirectly activate intracellular signaling pathways in HGFs [25]. These reports suggest that Clys and sIL-6R are also involved in the pathogenesis of peri-implantitis and need to be clarified. Therefore, in this study, the colonization rate of *Ca* and the level of sIL-6R in PICF of peri-implantitis sites were clinically determined, and the influence of Clys and sIL-6R in HGFs was clarified by this in vitro study. In a previous clinical study, the mPI and mGI were measured to evaluate the cleaning and inflammatory conditions of peri-implant tissues [26]. In addition, the level of IL-1β, a well-known inflammatory mediator, in the PICF of peri-implant inflammatory sites was compared with that in healthy sites [27].

At first, we examined a total of 36 implants from 19 patients (healthy and peri-implantitis subjects) who received oral implant therapy in the present clinical study. As shown in Figure 1, we found that the level of sIL-6R (1.48 ng/mL) in the PICF from the peri-implantitis group was significantly higher than that in the healthy group (0.44 ng/mL). sIL-6R binds to IL-6, and the IL-6/sIL-6R complex associates with gp130 on the cell membrane, which activates intracellular signaling through the JAK/STAT pathway. Theoretically, all cells contain gp130 and can be activated by the IL-6/sIL-6R complex, which has been reported to have a potentially dramatic cellular response [28]. Previous studies have shown higher levels of IL-6 in the PICF from peri-implantitis sites [27]. Given that IL-6 is a typical inflammation-related factor produced by gingival fibroblasts and other cells that construct peri-implant tissues, sIL-6R, its receptor, may be involved in the pathology of peri-implantitis via various inflammatory cascades. Therefore, in this study, the higher level of sIL-6R in PICF from peri-implantitis sites is considered to be indirectly involved in the pathogenesis of peri-implantitis. Bevilacqua et al. [29] reported that the volume of GCFs from periodontitis sites and PICF from peri-implantitis sites were comparable. Carcuac et al. [22] revealed that peri-implantitis lesions had significantly greater numbers and densities of lymphocytes and macrophages than periodontitis lesions, indicating histopathological differences between the two diseases. These previous studies suggest a difference between sIl-6R levels in the lesions of these two diseases. Therefore, in this study, regarding the use of sIL-6R as a possible biomarker for peri-implantitis, the involvement of sIL-6 in the pathogenesis of peri-implantitis was also investigated by in vitro experiments using HGFs.

In regard to *Ca* infection at implanted sites, the colonization rates of *Ca* in healthy sites and peri-implantitis sites were 11.1% and 33.3%, respectively, and the average numbers of colonies in healthy sites and peri-implantitis sites were 1.5 and 8.5, respectively. These results indicated that *Ca* in peri-implantitis sites tended to be detected more often and reside more often than in healthy sites. A systematic review of the implication of Candida in peri-implantitis reported that the percentage of Candida presence ranged from 3–76.7% [23]. These studies determined the presence of Candida using blood agar, CHROM agar, quantitative real-time PCR, and randomly amplified polymorphic DNA. In this clinical study, the number of patients was limited, and the culturing method was used; however, the results were similar to those of previous reports. However, it is not clear whether the distribution and localization of *Ca* colonized the pocket of peri-implantitis lesions, because we only evaluated the presence of *Ca* in the entire pocket. Moreover, no reports show the concentration level of Clys in the PICF or surrounding gingival tissue at the *Ca*-infected peri-implantitis site. These issues should be addressed in future studies.

Next, to determine the involvement of *Ca* in peri-implantitis, we conducted in vitro studies on the cellular response of HGFs to Clys derived from Ca. This is the first study on the effects of Clys on HGFs, which comprise the connective tissue of the peri-implant mucosa. For the reason that cytotoxicity was significant in HGFs stimulated with 10 μM Clys for 24 h (Figure 2), up to 5 μM Clys was used for subsequent experiments. The cytotoxicity of Clys in epithelial cells has been reported to be significant at 15 μM [12]. Westman et al. [30] discovered that epithelial cells repair mucosal damage caused by *Ca* through an Alg-2/Alix/ESCRT-III-dependent blebbing process and lysosomal membrane exocrine secretion. The cytotoxicity and tissue destruction of Clys may depend on the differences in the membrane structure and repair function of cells. 

IL-6 is a key pro-inflammatory cytokine produced by macrophages upon bacterial infection and is involved in bacterial elimination and tissue repair [28]. However, the continuous production of this cytokine has been implicated in the pathogenesis of various chronic inflammatory diseases. MMP-1 plays an important role in the destruction of type 1 collagen fibers in peri-implant tissues [31]. TIMPs are molecules that regulate the function of MMPs, and the balance between MMPs and TIMPs is essential for the maintenance of peri-implant mucosa [32]. Clys significantly produced IL-6 and MMP-1 in HGFs in a concentration-dependent manner (Figure 3). However, it did not induce TIMP-1, IL-8, or MCP-1 production. These results suggest that Clys plays a critical role in the pathogenesis of connective tissue destruction at peri-implantitis sites. Regarding the intracellular signaling pathways activated by Clys stimulation, Clys increased the production of IL-6 and pro-MMP-1 via the p38 and ERK pathways, respectively (Figure 4). These results suggest that the intracellular signaling pathways involved in IL-6 and MMP-1 production after Clys stimulation are different. However, detailed intracellular signaling pathways after Clys stimulation should be determined at various time points, and the function of their secreted products should be analyzed in further experiments. Clys-induced immune responses in epithelial cells have been reported to activate the EGFR signaling pathway via EGFR phosphorylation and EGFR ligand release, and MMPs are required for this signaling pathway activation [33]. Recently, it was discovered that Clys activates the EGFR-ERK pathway in epithelial cells, whereas p38 activation is regulated independently of EGFR [34]. Therefore, in addition to the Clys-induced activation of the MAPKs pathway reported in previous studies, a more complex intracellular signal activation mechanism may be present in HGFs. This study revealed that sIL-6R levels in PICF from peri-implantitis sites were significantly higher in clinical studies and that Clys induced IL-6 production in HGFs in vitro. HGFs do not express enough IL-6R on the cell surface to adequately bind IL-6 but do express gp130, which can bind to the IL-6/sIL-6R complex in the presence of sIL-6R, such as in periodontitis [35]. The present study demonstrated that the production levels of IL-6 and pro-MMP-1 in HGFs co-stimulated with Clys and sIL-6R were significantly higher than those in Clys-only stimulation. Interestingly, IL-8 production was also greatly increased in HGFs co-stimulated with Clys and sIL-6R. These results suggest that the IL-6/sIL-6R complex, consisting of IL-6 induced in Clys-stimulated HGFs and sIL-6R in peri-implantitis sites, exacerbates cellular inflammatory responses, leading to the destruction of peri-implant tissues. IL-6 produced within 48 h of Clys stimulation forms a complex with sIL-6R, and this complex significantly increased IL-8 and pro-MMP-1 in gingival fibroblasts at 48 h. These results suggest that IL-6 levels sufficient to form a complex with sIL-6R reached 48 h after stimulation with Clys. However, regarding the pathogenesis of peri-implantitis, it still remains unclear when sIL-6R is up-regulated during peri-implantitis. A previous study revealed that IL-6R is expressed in leukocyte and hepatocyte membranes, and the concentration of its soluble form, sIL-6R, shed from the neutrophil membrane is high in neutrophil-enriched inflammatory fluids [36]. Therefore, sIL-6R may be produced by immune cells, particularly neutrophils, in peri-implantitis lesions in the early or chronic stages. This possibility should be investigated further in in vitro and clinical studies.

This report raises the possibility that Clys of *Ca* infected at implanted sites triggers a cascade of events that initiate and exacerbate pro-inflammatory reactions leading to peri-implantitis. However, the effect of *Ca* on the severity of peri-implantitis is unclear. Therefore, it is necessary to determine the colonization rate of the hyphal form of *Ca*, which is more virulent than the yeast form, in the peri-implant sulcus of patients with peri-implantitis in further clinical studies. Previous studies have shown that *Ca* and *Porphyromonas gingivalis* form a community through InlJ-Als3-dependent binding and reported that *P. gingivalis* within the community may increase pathogenicity [37]. João et al. [38] reported that the interaction between *Ca* and bacteria involves the environment, the composition of the microbiota, and the host immune response and pointed out that most studies have focused on the interaction between *Ca* and individual oral bacterial species. However, few studies have investigated the interactions between *Ca* and polymicrobial biofilms. Therefore, further studies on the role of *Ca* in the microbial flora with regard to the severity of peri-implantitis are needed.

## 4. Materials and Methods

### 4.1. Clinical Study 

#### 4.1.1. Patients and Clinical Examinations

A total of 36 oral implants from 18 healthy sites and 18 peri-implantitis sites were examined in this cross-sectional study. Patients who visited the outpatient clinic of the Oral Implant Center at Tokushima University Hospital were enrolled in this study. Peri-implantitis was diagnosed based on the consensus report of the 2017 World Workshop on the Classification of Periodontal and Peri-Implant Diseases and Conditions. The pocket depth, mPI, and mGI were evaluated by trained dentists using a peri-implant probe (Hu-Friedy, COLORVUE^®^ PROBES 3-6-8-11, Tokyo, Japan). Exclusion criteria are as follows: 1. Pregnant women or lactating women. 2. Patients using anti-inflammatory drugs. 3. Patients with malignancy or immunodeficiency, or during radiotherapy or chemotherapy. 4. Patients with smoking. The study protocol was approved by the local ethics committee of Tokushima University Hospital (No. 3723), and the study was conducted in accordance with the principles of the Declaration of Helsinki. 

#### 4.1.2. Peri-Implant Crevicular Fluid (PICF) Sampling

After the removal of the supra-gingival plaque, sampling sites were isolated with cotton rolls and gently air-dried. PICF samples were obtained from healthy or peri-implantitis sites by inserting paper points (absorbent paper points #45, Dentsply/Maillefer, Ballaigues, Switzerland) and keeping them in place for 30 s, as described in a previous report [18]. Paper points containing PICF were applied firmly to CHROMagar and cultured at 37 °C for 48 h in an incubator, then evaluated for the presence of *Ca*, and colonies grown on agar plates were counted. In addition, the PICFs were collected from different peri-implant sites using the same protocol. The PICF samples were placed in sterile Eppendorf tubes containing 100 μL PBS with a protease inhibitor cocktail (Roche Diagnostics, Basel, Switzerland) on ice and the stored at −80 °C until use.

#### 4.1.3. The Measurement of sIL-6R and IL-1β Levels Using Enzyme-Linked Immunosorbent Assay (ELISA)

The levels of sIL-6R and IL-1β in the collected PICF were measured using a commercially available ELISA kit (R&D Systems, Minneapolis, MN, USA).

#### 4.1.4. Statistical Analysis

In the clinical study, statistical significance was determined using the Mann-Whitney U test and Pearson’s chi-square test for cross-sectional analysis. Statistical analyses were performed using SPSS version 25 (IBM Corp., Armonk, NY, USA). Statistical significance was set at *p* < 0.05.

### 4.2. In Vitro Study

#### 4.2.1. Reagents

Clys were obtained from PEPTIDE Inc. (Osaka, Japan). sIL-6R was purchased from R&D Systems. Antibodies against phospho-p44/42 MAPK, total-p44/42 MAPK, phospho-JNK MAPK, total-JNK MAPK, phospho-p38, total-p38 and β-actin were obtained from Cell Signaling Technology (Danvers, MA, USA). U0126 (mitogen-activated protein kinase kinase [MEK] inhibitor) and SB2013580 (p38 MAPK inhibitor) were purchased from Wako Pure Chemical Industries (Osaka, Japan).

#### 4.2.2. Cell Culture

Human gingival fibroblasts (HGFs) cell line CRL-2014^®^ (American Type Culture Collection; ATCC, Manassas, VA, USA) was cultured in Dulbecco’s modified Eagle’s medium (DMEM) and 10% fetal bovine serum (FBS, Biosera, Kansa City, MO, USA) until it reached subconfluence, as described in a previous report [39].

#### 4.2.3. Cell Cytotoxicity Assay

To examine the effects of Clys on cell cytotoxicity, an LDH assay was performed using a Cytotoxicity LDH assay kit (DOJINDO, Kumamoto, Japan). HGFs were cultured until subconfluence and treated with Clys (concentration: 0.1, 1, 2.5, 5, and 10 μM) for 24 or 48 h. The amount of formazan dye produced was measured at 490 nm using a microplate reader (Bio-Rad, Hercules, CA, USA).

#### 4.2.4. ELISA for Inflammation-Related Molecules Produced from HGFs

After HGFs were treated with Clys (0.1, 1, 2.5, and 5 μM) for 24 h, cell culture supernatants were collected and immediately stored at −80 °C until use. The concentrations of IL-6, IL-8, pro-MMP-1, TIMP-1, and MCP-1 were measured using commercially available enzyme-linked immunosorbent assay (ELISA) kits (R&D Systems).

#### 4.2.5. Activations of Clys Induced-Intracellular Signaling

After changing the medium to DMEM containing 0.5% FBS and culturing for 24 h, HGFs were treated with Clys (0.1, 1, 2.5, or 5 μM) for 30 min, and total cellular proteins were extracted using lysis buffer (0.5% SDS, 10 mM Tris-HCl pH7.4, and protease inhibitor mix: Complete^TM^ [Sigma-Aldrich, St. Louis, MO, USA]). Phosphorylation of ERK, JNK, and p-38 was examined using Western blotting. All Western blotting images are representative of three independent experiments. To investigate the potential pathways contributing to Clys-induced inflammation-related molecule production, HGFs were pretreated with U0126 (MEK inhibitor, 10 μM) and SB2013580 (p38 MAPK inhibitor, 10 μM) for 1 h and treated with Clys (5 μM) for 24 h, according to a previous study [39]. The levels of IL-6 and pro-MMP-1 in the cell culture supernatant were measured using commercially available ELISA kits (R&D Systems).

#### 4.2.6. Inflammation-Related Molecules Production via gp130/Stat3 Pathway in the Presence of sIL-6R

HGFs were stimulated with Clys (5 μM) for 24 or 48 h, and then sIL-6R (50 ng/mL) was applied for 30 min. Total cellular proteins were extracted using a lysis buffer. Activation of gp130 and Stat3 in HGFs was examined using Western blotting. After co-stimulation with Clys (5 μM) and sIL-6R (50 ng/mL), the levels of IL-6, pro-MMP-1, TIMP-1, and IL-8 in the cell culture supernatant were measured using commercially available ELISA kits (R&D Systems).

#### 4.2.7. Statistical Analysis

Statistical significance was determined by analysis of variance (ANOVA) and Tukey-honest significant difference analysis for the in vitro study. Statistical analyses were performed using SPSS version 25 (IBM Corp., Armonk, NY, USA). Statistical significance was set at *p* < 0.05.

## 5. Conclusions

Candidalysin, Clys, derived from *Ca*, increases IL-6 and pro-MMP-1 production in human gingival fibroblasts (HGFs) via ERK and p38 MAPK pathways. The complex formed with sIL-6R in peri-implantitis lesions and IL-6 produced from Clys-stimulated HGFs exacerbates the inflammatory response to produce IL-8, IL-6, and pro-MMP-1 and activate the gp130/STAT3 pathway. Collectively, Clys from *Ca* plays an important role in the pathogenesis of peri-implantitis through the induction of pro-inflammatory mediators.

## Figures and Tables

**Figure 1 ijms-24-03256-f001:**
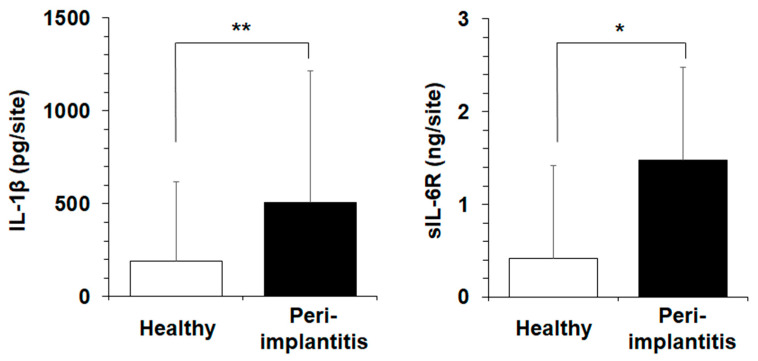
Inflammatory parameters in peri-implant crevicular fluid (PICF) of health and Peri-implantitis patients. Levels of interleukin (IL)-1β and soluble interleukin (sIL)-6R in the collected PICF were measured using an enzyme-linked immunosorbent assay (ELISA) kit. Data are the mean ± SD. Values of * *p* < 0.05 and ** *p* < 0.01 were considered to indicate statistical significance (Mann-Whitney U test).

**Figure 2 ijms-24-03256-f002:**
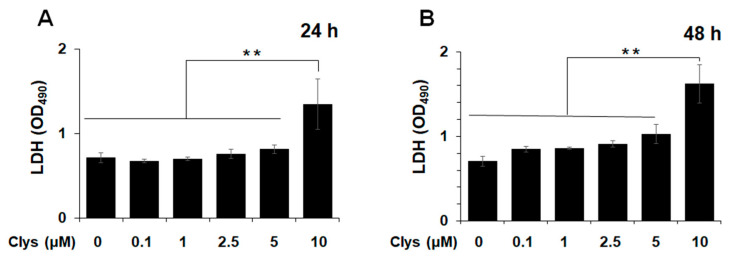
Cell cytotoxicity of Candidalysin (Clys) to human gingival fibroblasts (HGFs). After HGFs were treated with Clys (0.1, 1, 2.5, 5, 10 μM) for (**A**) 24 h and (**B**) 48 h, the cell cytotoxicity was determined by lactate dehydrogenase (LDH) assay. Values of ** *p*  <  0.01 was considered to indicate statistical significance (one-way analysis of variance with a post-hoc Tukey’s test).

**Figure 3 ijms-24-03256-f003:**
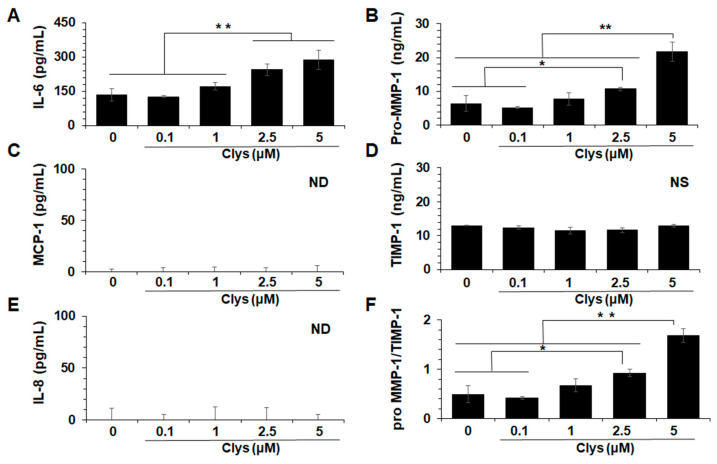
Production levels of inflammation-related molecules in human gingival fibroblasts (HGFs) stimulated with Candidalysin (Clys). After HGFs were treated with Clys (0.1, 1, 2.5, 5 μM) for 24 h, the production levels of inflammation-related molecules were quantified by enzyme-linked immunosorbent assay (ELISA). (**A**) IL-6, (**B**) pro-MMP-1, (**C**) MCP-1, (**D**) TIMP-1, (**E**) IL-8, (**F**) the ratio of pro-MMP-1/TIMP-1. Abbreviations: NS: Non significant, ND: Not detected. Values of ** *p*  <  0.01 and * *p*  <  0.05 were considered to indicate statistical significance (one-way analysis of variance with a post-hoc Tukey’s test).

**Figure 4 ijms-24-03256-f004:**
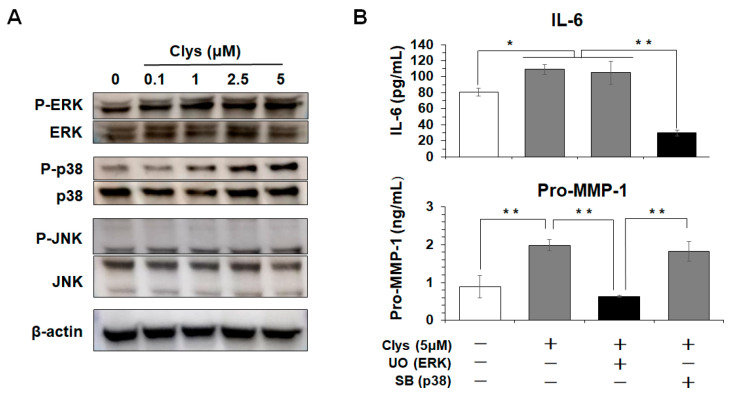
Candidalysisn (Clys)-induced intracellular signaling pathways, MAPKs, in human gingival fibroblasts (HGFs). (**A**) Phosphorylation levels of Clys-induced intracellular signaling pathways, MAPKs, in HGFs. HGFs were cultured with Clys (0.1, 1, 2.5, 5 μM) for 30 min. Cell lysates were separated by sodium dodecyl sulfate-polyacrylamide gel electrophoresis (SDS-PAGE) and analyzed using Western blotting probed with antibodies against phospho-ERK/total-ERK, phospho-p38/total-p38 and phospho- c-Jun N-terminal kinase (JNK)/total-JNK as well as β-actin. (**B**) HGFs were pre-treated with U0126 (10 μM), or SB2013580 (10 μM) for 1 h and then treated with 5 μM Clys for 24 h. Production levels of IL-6 and pro-MMP-1 in HGFs were quantified using ELISA. The production levels of IL-6 and pro-MMP-1 were significantly decreased by the p38-specific inhibitor SB2013580 and the ERK-specific inhibitor U0126, respectively (Black column). Values of ** *p*  < 0.01 and * *p*  < 0.05 were considered to indicate statistical significance (one-way analysis of variance with a post-hoc Tukey’s test).

**Figure 5 ijms-24-03256-f005:**
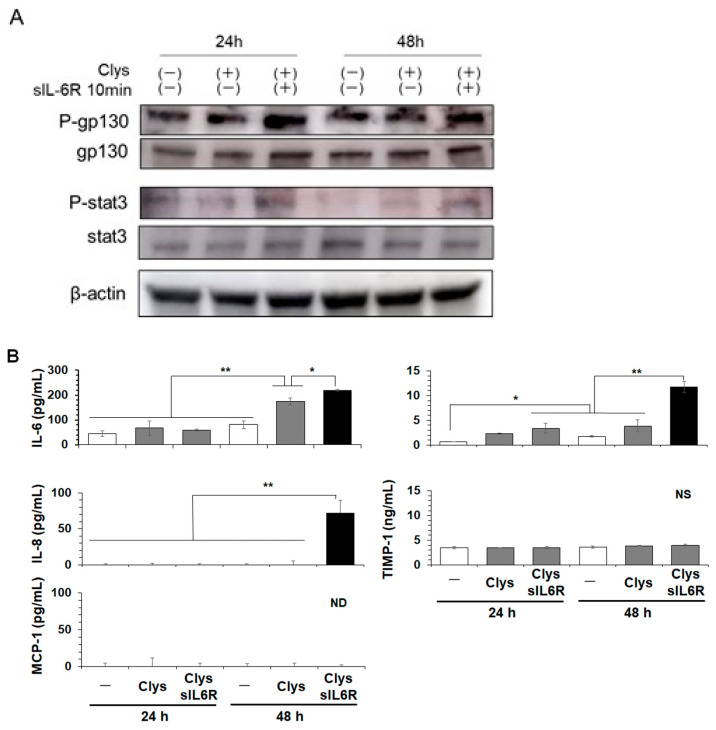
Effects of soluble interleukin (sIL)-6R on Candidalysisn (Clys)-induced inflammation-related molecules production via gp130/Stat3. (**A**) Phosphorylation levels of intracellular signaling pathways, gp130 and Stat3, in Clys (5 μM) and sIL-6R (50 ng/mL)-stimulated human gingival fibroblasts (HGFs). HGFs were stimulated with Clys (5 μM) for 24 or 48 h, and then sIL-6R (50 ng/mL) was applied for 30 min. Cell lysates were separated by sodium dodecyl sulfate-polyacrylamide gel electrophoresis (SDS-PAGE) and analyzed using Western blotting probed with antibodies against phospho-gp130/total-gp130 and phospho-Stat3/total-Stat3. (**B**) After co-stimulation with Clys (5 μM) and sIL-6R (50 ng/mL), production levels of IL-6, pro-MMP-1, IL-8, TIMP-1 and MCP-1 in HGFs were quantified using ELISA. The production levels of IL-6, pro-MMP-1, and IL-8 at 48 h in HGFs co-stimulated with Clys and sIL-6R (Black columns) were significantly higher than those in HGFs stimulated with Clys only. Values of ** *p* < 0.01 and * *p* < 0.05 were considered to indicate statistical significance (one-way analysis of variance with a post-hoc Tukey’s test).

**Table 1 ijms-24-03256-t001:** Demographic parameters and clinical indices of health and peri-implantitis patients.

Parameters	Patient Who Receives Implant Treatment		*p*-Value
	Healthy	Peri-Implantitis	
Number of subjects (M/F)	9 (2/7)	10 (3/7)	0.70
Age (yr)	63.44 ± 8.22	69.2 ± 14.75	0.36
Number of implant	18	18	
PPD	2.72 ± 0.57	5.72 ± 1.45	0.001 **
mPI	0.33 ± 0.49	2.11 ± 0.68	0.001 **
mGI	0.78 ± 0.94	2.0 ± 0.77	0.001 **
Number of *C. albicans*-positive subjects	2	6	0.199
Colonization rate (%)	11.1	33.3	
Average number of colonies	1.5	8.5	0.143

Abbreviations: PPD, probing pocket depth; mPI, modified plaque index; mGI, modified gingival index. A value of ** *p*  < 0.01 was considered to indicate statistical significance (Pearson’s chi-square test, Mann-Whitney U test).

## Data Availability

The data presented in this study are available on request from the corresponding authors.

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
