# Peer review of "Effects of Candidalysin Derived from Candida albicans on the Expression of Pro-Inflammatory Mediators in Human Gingival Fibroblasts"

_ijms, 2023, doi:10.3390/ijms24043256_

Round 1

Reviewer 1 Report (Previous Reviewer 1)

Comments to the authors:

The authors showed that candidalysin from Candida albicans enhances proinflammatory responses in human gingival fibroblasts and may participate in peri-implantitis. Although the topic is very interesting, there is some background information and data based discussion missing.

Major concerns:

  1. What is the difference between periodontitis and peri-implantitis pathologically? Please discuss further. Is Candida albicans also associated with periodontitis pathogenesis?
  2. Generally in the oral cavity, if Candida albicans is recognized as a major pathogen, the microbiome is dominated by it. The authors showed that this is not the case. Do you think Candida albicans may have interactions with some microbacteria for its pathogenesis, or other possibilities for that? Please discuss.
  3. Where was Candida albicans colonized in peri-implantitis case?
  4. Is Candidalysin detected in PICF or gingival tissue? If so, what is the concentration?
  5. Is candida albicans presence or colonization rate related to peri-implantitis severity?
  6. Are all the western blot images representative images from 3-5 experiments? If so, please describe in the material and methods section.
  7. In figure 4A, ERK and p38 are already phosphorylated before Clys stimulation. Did you do starvation without FBS before the stimulation? Please clarify in the material and methods section.
  8. Figure 4 indicates that IL-6 and Pro-MMP-1 do not share the same pathway. Do you think this is because of the time point when they get secreted or its functions? Please discuss.
  9. In figure 5A, Stat3 protein amount also increased in 24h. Please show a better image for Stat3 blot.
  10. In figure 5B, the way Clys single or Clys+sIL6R double stimulation affects on each cytokine’s production is very different (IL-6 is similar, IL-8 and Pro-MMP-1 are increased). Do you have any thoughts about the reason? When sIL6R expression comes up during peri-implantitis pathogenesis?   

Author Response

February 02, 2023

Prof. Dr. Robert Ancuceanu

Faculty of Pharmacy,

Carol Davila University of Medicine and Pharmacy,

020956 Bucharest, Romania

Special Issue Editor

Special Issue “Fungal Molecular Mechanisms, Fungal Infections and Antifungal Drugs”

in International Journal of Molecular Sciences

Dear Prof. Robert Ancuceanu:

     I would like to submit the revised manuscript entitled “Effects of Candidalysin derived from Candida albicans on the expression of pro-inflammatory mediators in human gingival fibroblasts” (MS ID: ijms-2183967) for publication in International Journal of Molecular Sciences, Special Issue “Fungal Molecular Mechanisms, Fungal Infections and Antifungal Drugs”.  We have organized the manuscript after addressing the comments and points raised by the reviewers and the editor.

     Please find the enclosed revised manuscript as well as the response letter to the comments of the editor and the reviewers in the following file.  I look forward to your review of the revised manuscript and thank you for considering it for publication in International Journal of Molecular Sciences, Special Issue “Fungal Molecular Mechanisms, Fungal Infections and Antifungal Drugs”.

Sincerely,

Hiromichi Yumoto, Ph.D., D.D.S.

Professor

Department of Periodontology and Endodontology

Institute of Biomedical Sciences,

Tokushima University Graduate School

3-18-15 Kuramoto-cho, Tokushima, 770-8504, JAPAN

Tel.: +81-88-633-7343

Fax: +81-88-633-7345

E-mail: yumoto@tokushima-u.ac.jp

Response to the Reviewer’s and Editor’s Comments

Reviewer #1

  1. What is the difference between periodontitis and peri-implantitis pathologically? Please discuss further. Is Candida albicans also associated with periodontitis pathogenesis?

We wish to thank the reviewer for the useful comments. Regarding the pathological difference between periodontitis and peri-implantitis and the association between Candida albicans and pathogenesis of periodontitis, the following sentences were described in the revised both “Introduction” and “Discussion” sections.

(In the Introduction section) (Lines 52 – 55)

A previous systematic review reported that the microbiological characteristics of peri-implantitis and periodontitis are different and that the quantity rather than the quality of the microorganisms, is the determinant in the development of peri-implantitis [6].

(In the Discussion section)

(Lines 192 – 196)

Clinical features of peri-implantitis and periodontitis are common; however, significant histopathological differences exist between these lesions [21]. Previous studies have shown that peri-implantitis lesions are more than twice as large as periodontitis lesions and have significantly greater proportions, numbers, and densities of immune cells [22].

(Lines 203 - 205)

Canabarro et al. [24] reported that subgingival colony formation in some yeasts, particularly Ca, is associated with the severity of chronic periodontitis.

  1. Generally in the oral cavity, if Candida albicans is recognized as a major pathogen, the microbiome is dominated by it. The authors showed that this is not the case. Do you think Candida albicans may have interactions with some microbacteria for its pathogenesis, or other possibilities for that? Please discuss.

We wish to thank the reviewer for the useful comments. Regarding the roles of Candida albicans among oral microbiome and its interactions with some microorganism on the pathogenesis of peri-implantitis, the following sentences were described in the revised “Discussion” sections.

(In the Discussion section) (Lines 312 – 320)

Previous studies have shown that Ca and Porphyromonas gingivalis form a com-munity through InlJ-Als3-dependent binding and have reported that P. gingivalis within the community may increase pathogenicity [37]. João et al. [38] reported that the interaction between Ca and bacteria involves the environment, composition of the microbiota, and the host immune response, and pointed out that most studies have focused on the interaction between Ca and individual oral bacterial species. However, few studies have investigated the interactions between Ca and polymicrobial biofilms. Therefore, further studies on the role of Ca in the microbial flora with regard to the severity of peri-implantitis are needed.

  1. Where was Candida albicans colonized in peri-implantitis case?

We wish to thank the reviewer for the useful comments. The distribution and localization of C. albicans colonized in the pocket of peri-implantitis lesion remain unclear in this study. Therefore, this issue should be determined in further study. The following sentences were described in the revised “Discussion” sections.

(In the Discussion section) (Lines 249 – 253)

However, it is not clear whether the distribution and localization of Ca colonized the pocket of peri-implantitis lesions, because we only evaluated the presence of Ca in the entire pocket. Moreover, no reports are showing the concentration level of Clys in the PICF or surrounding gingival tissue at the Ca-infected peri-implantitis site. These issues should be addressed in future studies.

  1. Is Candidalysin detected in PICF or gingival tissue? If so, what is the concentration?

We wish to thank the reviewer for the useful comments. There are no report showing the concentration level of Clys in PICF or surrounding gingival tissue at C. albicans-infected peri-implantitis site. Therefore, this issue should be determined in further study. The following sentences were described in the revised “Discussion” sections.

(In the Discussion section) (Lines 249 – 253)

However, it is not clear whether the distribution and localization of Ca colonized the pocket of peri-implantitis lesions, because we only evaluated the presence of Ca in the entire pocket. Moreover, no reports are showing the concentration level of Clys in the PICF or surrounding gingival tissue at the Ca-infected peri-implantitis site. These issues should be addressed in future studies.

  1. Is candida albicans presence or colonization rate related to peri-implantitis severity?

We wish to thank the reviewer for the useful comments. However, the influence of this fungus, C. albicans, on the severity of peri-implantitis is unclear. Therefore, regarding this issue, it is required to determine the colonization rate of hyphal form of C. albicans, which is more virulent than the yeast form, in the peri-implant sulcus of patients with peri-implantitis in further clinical study. The following sentences were described in the revised “Discussion” sections.

(In the Discussion section) (Lines 304 – 307)

However, the effect of Ca, on the severity of peri-implantitis is unclear. Therefore, it is necessary to determine the colonization rate of the hyphal form of Ca, which is more virulent than the yeast form, in the peri-implant sulcus of patients with peri-implantitis in further clinical studies.

  1. Are all the western blot images representative images from 3-5 experiments? If so, please describe in the material and methods section.

We wish to thank the reviewer for the useful comments. All western blot images are representative images from three independent experiments. The following sentence was described in the revised “Materials and Methods” section.

(In the Materials and Methods section) (Lines 385 – 386)

All Western blotting images are representative of three independent experiments.

  1. In figure 4A, ERK and p38 are already phosphorylated before Clys stimulation. Did you do starvation without FBS before the stimulation? Please clarify in the material and methods section.

We wish to thank the reviewer for the useful comments. We changed the medium to DMEM containing 0.5% FBS and cultured HGFs for 24 hrs before the treatment with Clys (0.1, 1, 2.5, or 5 M) for 30 min. The following modified sentence was described in the revised “Materials and Methods” section.

(In the Materials and Methods section) (Lines 381 – 384)

After changing the medium to DMEM containing 0.5% FBS and culturing for 24 h, HGFs were treated with Clys (0.1, 1, 2.5, or 5 M) for 30 min, and total cellular proteins were extracted using lysis buffer (0.5% SDS, 10 mM Tris-HCl pH7.4, and protease inhibitor mix: CompleteTM [Sigma-Aldrich, St. Louis, MO, USA]).

  1. Figure 4 indicates that IL-6 and Pro-MMP-1 do not share the same pathway. Do you think this is because of the time point when they get secreted or its functions? Please discuss.

We wish to thank the reviewer for the useful comments. Regarding the intracellular signaling pathways after Clys stimulation, the following sentences were described in the revised “Discussion” sections.

(In the Discussion section) (Lines 275 – 279)

These results suggest that the intracellular signaling pathways involved in IL-6 and MMP-1 production after Clys stimulation are different. However, detailed intracellular signaling pathways after Clys stimulation should be determined at various time points and the function of their secreted products should be analyzed in further experiments.

  1. In figure 5A, Stat3 protein amount also increased in 24h. Please show a better image for Stat3 blot.

We wish to thank the reviewer for the useful comments. As suggested by the reviewer, the image of stat3 from western blot shown in Figure 5A was modified in the revised manuscript.

  1. In figure 5B, the way Clys single or Clys+sIL6R double stimulation affects on each cytokine’s production is very different (IL-6 is similar, IL-8 and Pro-MMP-1 are increased). Do you have any thoughts about the reason? When sIL6R expression comes up during peri-implantitis pathogenesis?   

We wish to thank the reviewer for the useful comments. Regarding the effects of co-stimulation with Clys and sIL6R as well as the sIL-6 expression on the pathogenesis of peri-implantitis, the following sentences were described in the revised “Discussion” sections.

(In the Discussion section) (Lines 296 – 306)

IL-6 produced within 48 h of Clys stimulation forms a complex with sIL-6R and this complex significantly increased IL-8 and pro-MMP-1 in gingival fibroblasts at 48 h. These results suggest that IL-6 levels sufficient to form a complex with sIL-6R reached 48 h after stimulation with Clys. However, regarding the pathogenesis of peri-implantitis, it still remains unclear when is sIL-6R up-regulated during peri-implantitis. A previous study revealed that IL-6R is expressed in leukocyte and hepatocyte membranes, and the concentration of its soluble form, sIL-6R, shed from the neutrophil membrane is high in neutrophil-enriched inflammatory fluids [36]. Therefore, sIL-6R may be produced by immune cells, particularly neutrophils, in peri-implantitis lesions at the early or chronic stages. This possibility should be investigated in further in vitro and clinical studies.

Reviewer 2 Report (New Reviewer)

The manuscript titled “Effects of Candidalysin derived from Candida albicans on the expression of pro-inflammatory mediators in human gingival fibroblasts” aimed to clarify Candida albicans prevalence in peri-implant sulcus and to investigate the effects of a toxin produced by C. albicans (candidalysin), on human gingival fibroblasts.

The manuscript is interesting and quite-well written.

Introduction:

Line 34: correct the sentence: add “been” (has already been established as a useful method…) method of what?

Line 49: correct: “has been occasionally caused”?

In general, this section could be enhanced with more epidemiological data

https://doi.org/10.1186/s12903-022-02493-8

https://www.mdpi.com/1660-4601/19/19/12667

Please correct the abbreviation for Candidalysin to Clys.

The aim of study (line 80-83) should be revised.

Results:

In Table 1, in what units "C. albicans presence" is presented?

Line 110, please explain abbreviations at their first use.

In discussion, please relate to the clinical significance of the study findings

Minor issues:

Check English grammar.

Please put the reference number right after the author’s name of the cited study

Author Response

February 02, 2023

Prof. Dr. Robert Ancuceanu

Faculty of Pharmacy,

Carol Davila University of Medicine and Pharmacy,

020956 Bucharest, Romania

Special Issue Editor

Special Issue “Fungal Molecular Mechanisms, Fungal Infections and Antifungal Drugs”

in International Journal of Molecular Sciences

Dear Prof. Robert Ancuceanu:

     I would like to submit the revised manuscript entitled “Effects of Candidalysin derived from Candida albicans on the expression of pro-inflammatory mediators in human gingival fibroblasts” (MS ID: ijms-2183967) for publication in International Journal of Molecular Sciences, Special Issue “Fungal Molecular Mechanisms, Fungal Infections and Antifungal Drugs”.  We have organized the manuscript after addressing the comments and points raised by the reviewers and the editor.

     Please find the enclosed revised manuscript as well as the response letter to the comments of the editor and the reviewers in the following file.  I look forward to your review of the revised manuscript and thank you for considering it for publication in International Journal of Molecular Sciences, Special Issue “Fungal Molecular Mechanisms, Fungal Infections and Antifungal Drugs”.

Sincerely,

Hiromichi Yumoto, Ph.D., D.D.S.

Professor

Department of Periodontology and Endodontology

Institute of Biomedical Sciences,

Tokushima University Graduate School

3-18-15 Kuramoto-cho, Tokushima, 770-8504, JAPAN

Tel.: +81-88-633-7343

Fax: +81-88-633-7345

E-mail: yumoto@tokushima-u.ac.jp

Response to the Reviewer’s and Editor’s Comments

Reviewer #2

Introduction:

・Line 34: correct the sentence: add “been” (has already been established as a useful method…) method of what?

We wish to thank the reviewer for the useful comments. As suggested by the reviewer, we have corrected the sentence and clearly mentioned oral implant therapy in the revised “Introduction” section. The following sentences were described in the revised “Introduction” section.

(In the Introduction section) (Lines 35 – 37)

Oral implant therapy has already been established as a useful prosthetic treatment to restore dentition and reestablish the masticatory function of partially or fully edentulous patients with high predictability and survival rates [1].

・Line 49: correct: “has been occasionally caused”?

We wish to thank the reviewer for the useful comments. We have modified the manuscript following the reviewer's suggestions. The following sentences were described in the revised “Introduction” section.

(In the Introduction section) (Lines 42 – 44)

However, it has been reported that peri-implant diseases, including peri-implant mucositis and peri-implantitis are caused at a certain rate as a biological complications after oral implant therapy.

・In general, this section could be enhanced with more epidemiological data

We wish to thank the reviewer for the useful comments. The following recent systematic review and meta-analysis showing the prevalence of peri-implantitis has been added in the revised “Introduction” section.

Pedro Diaz; Esther Gonzalo; Luis J. Gil Villagra; Barbara Miegimolle & Maria J. Suarez. What is the prevalence of peri-implantitis? A systematic review and meta-analysis. BMC Oral Health. 2022, 22, 449.

(In the Introduction section) (Lines 47 – 49)

A recent systematic review and meta-analysis reported that the prevalence of peri-implantitis was 19.53% at the patient level and 12.53% at the implant level [5].

・Please correct the abbreviation for Candidalysin to Clys.

We wish to thank the reviewer for the useful comments. As suggested by the reviewer, we have corrected the abbreviation for Candidalysin to Clys in the revised manuscript.

・The aim of study (line 80-83) should be revised.

We wish to thank the reviewer for the useful comments. As suggested by the reviewer, we have revised to paragraph appropriately.

・Results: In Table 1, in what units "C. albicans presence" is presented?

We wish to thank the reviewer for the useful comments. Regarding the unit of C. albicans presence, we have clearly mentioned as “number of C.albicans-positive subjects” in revised Table 1.

・Line 110, please explain abbreviations at their first use.

We wish to thank the reviewer for the useful comments. As suggested by the reviewer, we have spelled out “lactate dehydrogenase” for “LDH” in the revised manuscript.

・In discussion, please relate to the clinical significance of the study findings

We wish to thank the reviewer for the useful comments. As suggested by the reviewer, we have put the following sentences explaining the clinical significance of high level of sIL-6R at peri-implantitis sites in the revised "Discussion" section.

(In the Discussion section) (Lines 226 – 231)

Given that IL-6 is a typical inflammation-related factor produced by gingival fibroblasts and other cells that construct peri-implant tissues, sIL-6R, its receptor, may be involved in the pathology of peri-implantitis via various inflammatory cascades. Therefore, in this study, the higher level of sIL-6R in PICF from peri-implantitis sites is considered to be indirectly involved in the pathogenesis of peri-implantitis.

・Minor issues:

・Check English grammar.

We wish to thank the reviewer for the useful comments. The revised version of our manuscript has been checked by English language editing service as described in the revised “Acknowledgements” section.

(In the Acknowledgements section) (Lines 426 – 427)

We would like to thank Editage (www.editage.com) for English language editing.

・Please put the reference number right after the author’s name of the cited study

We wish to thank the reviewer for the useful comments. We have modified the issue suggested by the reviewer throughout the manuscript.

This manuscript is a resubmission of an earlier submission. The following is a list of the peer review reports and author responses from that submission.

Round 1

Reviewer 1 Report

Comments to the authors:

The authors showed that candidalysin from Candida albicans enhances proinflammatory responses in human gingival fibroblasts and may participate in peri-implantitis. Although the topic is very interesting, there is some background information and data based discussion missing.

Major concerns:

  1. What is the difference between periodontitis and peri-implantitis pathologically? Please discuss further. Is Candida albicans also associated with periodontitis pathogenesis?
  2. Generally in the oral cavity, if Candida albicans is recognized as a major pathogen, the microbiome is dominated by it. The authors showed that this is not the case. Do you think Candida albicans may have interactions with some microbacteria for its pathogenesis, or other possibilities for that? Please discuss.
  3. Where was Candida albicans colonized in peri-implantitis case?
  4. Is Candidalysin detected in PICF or gingival tissue? If so, what is the concentration?
  5. Is candida albicans presence or colonization rate related to peri-implantitis severity?
  6. Are all the western blot images representative images from 3-5 experiments? If so, please describe in the material and methods section.
  7. In figure 4A, ERK and p38 are already phosphorylated before Clys stimulation. Did you do starvation without FBS before the stimulation? Please clarify in the material and methods section.
  8. Figure 4 indicates that IL-6 and Pro-MMP-1 do not share the same pathway. Do you think this is because of the time point when they get secreted or its functions? Please discuss.
  9. In figure 5A, Stat3 protein amount also increased in 24h. Please show a better image for Stat3 blot.
  10. In figure 5B, the way Clys single or Clys+sIL6R double stimulation affects on each cytokine’s production is very different (IL-6 is similar, IL-8 and Pro-MMP-1 are increased). Do you have any thoughts about the reason? When sIL6R expression comes up during peri-implantitis pathogenesis?